# The Impact of Abnormal Lipid Metabolism on the Occurrence Risk of Idiopathic Pulmonary Arterial Hypertension

**DOI:** 10.3390/ijms241814280

**Published:** 2023-09-19

**Authors:** Yaqin Wei, Hui Zhao, Bill Kalionis, Xu Huai, Xiaoyi Hu, Wenhui Wu, Rong Jiang, Sugang Gong, Lan Wang, Jinming Liu, Shijin Xia, Ping Yuan, Qinhua Zhao

**Affiliations:** 1Department of Cardio-Pulmonary Circulation, Shanghai Pulmonary Hospital, Tongji University School of Medicine, Shanghai 200433, China; 21211280007@m.fudan.edu.cn (Y.W.); huizhao_usst@126.com (H.Z.); huaixuok@163.com (X.H.); hxydoctor@163.com (X.H.); wenhui5621006@126.com (W.W.); listening39@tongji.edu.cn (R.J.); gongsugang@tongji.edu.cn (S.G.); lanwang@tongji.edu.cn (L.W.); jinmingliu@tongji.edu.cn (J.L.); 2Department of Geriatrics, Shanghai Institute of Geriatrics, Huadong Hospital, Fudan University, Shanghai 200040, China; xiashijinhd@163.com; 3Institute of Bismuth Science, University of Shanghai for Science and Technology, Shanghai 200093, China; 4Department of Maternal-Fetal Medicine Pregnancy Research Centre, Royal Women’s Hospital, Parkville 3052, Australia; bill.kalionis@thewomens.org.au

**Keywords:** idiopathic pulmonary arterial hypertension, biomarker, free fatty acid, diagnose, lipid metabolism

## Abstract

The aim was to determine whether lipid molecules can be used as potential biomarkers for idiopathic pulmonary arterial hypertension (IPAH), providing important reference value for early diagnosis and treatment. Liquid chromatography–mass spectrometry-based lipidomic assays allow for the simultaneous detection of a large number of lipids. In this study, lipid profiling was performed on plasma samples from 69 IPAH patients and 30 healthy controls to compare the levels of lipid molecules in the 2 groups of patients, and Cox regression analysis was used to identify meaningful metrics, along with receiver operator characteristic curves to assess the ability of the lipid molecules to predict the risk of disease in patients. Among the 14 lipid subclasses tested, 12 lipid levels were significantly higher in IPAH patients than in healthy controls. Free fatty acids (FFA) and monoacylglycerol (MAG) were significantly different between IPAH patients and healthy controls. Logistic regression analysis showed that FFA (OR: 1.239, 95%CI: 1.101, 1.394, *p* < 0.0001) and MAG (OR: 3.711, 95%CI: 2.214, 6.221, *p* < 0.001) were independent predictors of IPAH development. Among the lipid subclasses, FFA and MAG have potential as biomarkers for predicting the pathogenesis of IPAH, which may improve the early diagnosis of IPAH.

## 1. Introduction

According to the latest “2022 ESC/ERS Guidelines for the diagnosis and treatment of pulmonary hypertension” [1], PAH is categorized into five types, of which IPAH is one of the most common types and belongs to the first category of PAH. The pathogenesis of PAH involves genetics, inflammation, immunity, metabolism, and other aspects, which also leads to the complexity and diversity of PAH, and the corresponding diagnostic and treatment options also differ [2,3]. Idiopathic pulmonary arterial hypertension (IPAH) is a progressive disease that impacts the precapillary pulmonary vasculature, but the specific risk factors that contribute to IPAH remain unknown [4]. IPAH causes elevated cardiac afterload, which can eventuate in right heart failure and mortality. Despite recent advances in therapies that target the pulmonary vasculature, IPAH continues to be a life-threatening disease, with newly diagnosed patients having a 3-year survival rate of approximately 60% [5,6]. However, the initial clinical manifestations of IPAH are non-specific and difficult to diagnose, which often leads to patients missing the best treatment time. While current targeted drugs for IPAH have notably enhanced the survival and quality of life for some patients, a considerable portion of patients do not benefit from these drugs, and some experience a poor prognosis with no significant improvement in their quality of life [4]. Therefore, more accurate and specific biomarkers are needed to improve early screening and diagnosis rates of patients, which will be clinically important in improving the prognosis of IPAH patients.

Lipid molecules are important biomolecules and are the most abundant substance in plasma, where the lipid in cell membranes accounts for 50% of its weight [7]. A large number of lipids are present in endoplasmic reticulum, Golgi apparatus, mitochondria and lysosomes [8]. The diverse structure of lipids contributes to their significant biological functions. These functions include a crucial role in regulating various life processes such as cell growth and differentiation, apoptosis, energy conversion between cells and tissues, material transport, information recognition, and signal transmission [9,10]. As a result, changes in lipid metabolism and lipid actions significantly impact the physiological functions of cells, which contributes to the development of pathological disorders in organisms [11]. Altered lipid content is frequently linked to metabolic disease, cardiovascular disease, tumor formation, and neurological disease. Recent research reveals alterations in lipid content correlates with abnormalities in the levels, activities, and gene expression patterns of multiple enzymes, which contributes to the progression of various diseases [9,12]. However, few reports investigate the relationship between lipid metabolism and IPAH. Earlier studies noted a significant reduction in plasma HDL-C levels among IPAH patients [13,14], with lower levels of HDL-C correlating with higher mortality rates in these patients [15]. These reports combined with information from other studies on lipid metabolism in cardiovascular diseases [15,16,17], led us to speculate that other abnormal lipids may also contribute to the progression of IPAH, and that specific lipids, or types of lipids, may be useful in detecting the onset of IPAH.

The objective of this study was to determine the distribution and level of lipid molecules in the plasma of individuals with IPAH and healthy individuals using liquid chromatography–mass spectrometry. Additionally, this study highlighted the potential of lipid molecules as biomarkers to predict IPAH.

## 2. Results

### 2.1. Population Characteristics

We screened healthy people matched with age and sex from the physical examination center as controls. Table 1 showed no significant differences (*p* ≥ 0.05) in age, sex and body mass index between the patients and the control group. The proportion of IPAH patients with WHO PAH functional classification (WHO-FC) grade III–IV was 59.4%, and the 6 min walking distance was (393.9 ± 104.8) m. The hemodynamics, biochemical indexes and targeted drug therapy of IPAH patients are listed in Table 1.

### 2.2. Differences in Lipid Content between IPAH Patients and Healthy Controls

To screen out the differences between the two groups, the orthogonal partial least squares discrimination analysis (PLS-DA) model was used for supervised multidimensional statistical analysis of the original data matrix rows. The further away the sample distribution point, the greater the difference. As shown in Figure 1A, sample distribution points of the test IPAH group and the healthy control group were distinguishable, indicating that there was indeed a difference in lipid content between the IPAH patient and the healthy person. The response permutation test (RPT) evaluated the model to ensure that there was no overfitting. As illustrated in Figure 1B, this study distinguished the differential levels of lipids between the IPAH group and the healthy control group using a computational approach.

#### 2.2.1. The 14 Lipid Subclasses in IPAH Patients and Healthy Controls

A total of 588 lipid species were detected by the liquid chromatography–mass spectrometer (LC–MS), which were classified into 14 subgroups. The triacylglycerol (TAG) group contained the largest number of lipid molecules, while very few lipid molecules of the sphingosine (SS) and sterol (St) lipid groups were detected (Figure 1C).

#### 2.2.2. Levels of the 14 Lipid Subclasses in IPAH Patients and Healthy Controls

By comparing the levels of the 14 lipid subclasses in IPAH patients and the healthy control group, we found that the levels of FFA, MAG, diacylglycerol (DAG), TAG, phosphatidic acid (PA), phosphatidyl ethanolamine (PE), SS and St in IPAH patients were a significantly higher compared to those in healthy controls (*p* < 0.001). The levels of LPS, PG, PI and PS also showed an increasing trend in patients (*p* < 0.05). LPA and SM were similar in both groups. Figure 2A–N shows the distribution of each lipid subclass between IPAH patients and healthy controls.

### 2.3. Levels of Lipid Subclasses and FFA between Males and Females

Studies show that the incidence of IPAH is strongly related to gender [18,19,20]. This study also explored whether there were differences in the levels of these lipid molecules between males and females (Table 2). Surprisingly, there were no significant differences in lipid levels between males and females in either the healthy control or IPAH patient groups. However, this study found statistically significant differences in eight lipid molecules between healthy control males and male IPAH patients, and in ten lipid molecules between healthy control women and female IPAH patients. However, the differential lipids between males and females were not completely consistent between the two subgroups. For example, TAG and St showed significant differences only among males, and the lipid level of male IPAH patients was higher than that of healthy control males. LPS, PG, PI and PS were significantly different only among female IPAH patients, and the level of LPS in female IPAH patients was higher than that in healthy control females. FFA, MAG, DAG, PA, PE and SS were significantly different between males and females IPAH patients, and in each case the levels in IPAH patients were higher than healthy controls. This study found no significant statistical difference in LPA and SM between male and female subgroups. These results suggested that sex differences exist in the levels of some lipid molecules.

### 2.4. Logistic Regression Analysis of IPAH Occurrence of Different Types of Lipids

We chose to use the incidence of IPAH as the dependent variable in the logistic regression analysis, and included the lipid level data of healthy controls and IPAH patients. The influence of lipid subtypes alone on the risk of IPAH was determined. The indicators with statistical significance in the univariate logistic regression analysis were also included in the multivariate logistic regression analysis. The indicators with the most independent predictive value were then selected. Figure 3 displays the results of univariate regression analysis where FFA, MAG, DAG, PE, and SS had a significant predictive value for the risk of developing IPAH. LPS, PA, PG, PI, PS, and sterols (St) also had a significant predictive value for the risk of developing IPAH. This study found that higher levels of certain lipid subclasses had a significant predictive value for the risk of developing IPAH. The highest odds ratio (OR) value was found for MAG. An increase in the levels of the MAG subclass was associated with a higher risk of developing IPAH (OR: 3.711, 95%CI: 2.214–6.221, *p* < 0.0001).

However, the metabolic mechanism of the human body is complex and changeable, and there may have been interactions between various lipids. Therefore, we performed multi-factor logistic regression, that included adjustment for age, sex and BMI, and the final results showed that FFA (OR: 1.208, 95%CI: 1.509–1.378, *p* < 0.01), MAG (OR: 3.494, 95%CI: 2.023–6.034, *p* < 0.0001) were independent predictors of IPAH risk.

### 2.5. ROC Analysis of FFA, MAG and Their Combined Detection to Predict IPAH

To further assess the predictive effect of FFA and MAG on the risk of IPAH, we performed ROC analysis and evaluated their predictive effect with the aid of AUC. As shown in Figure 4, the AUC of FFA is 0.789, the AUC of MAG is 0.862 and the AUC of FFA and MAG is 0.851. Next, according to the highest Youden index, the cut-off values with the highest sensitivity and specificity of FFA, MAG and FFA and MAG were selected to define the high and low levels of FFA, MAG and FFA and MAG.

To further validate the predictive value of these two lipids on the incidence of IPAH, we defined the optimal cut-off values of FFA and MAG as “high” or “low”. We defined the optimal cut-off values of FFA and MAG as “high level” or “low level”. Then, we grouped the lipid level data of 69 IPAH patients and 30 controls, after which we calculated the proportion of IPAH patients in each group. 

The results of the chi-square test were statistically significant in each group (*p* < 0.001). The percentage of IPAH patients in the “high FFA level” group (85.94%) was higher than that in the “low FFA level” group (Figure 5A). Similarly, the percentage of IPAH patients in the “high MAG level” group was 93.33%, which was higher than that in the “low MAG level” group (Figure 5B). When the two lipids were evaluated together, the group with high levels of FFA and MAG had a 100% diagnosis rate of IPAH patients. When both FFA and MAG levels were low, patients had the lowest diagnosis rate of 10.53% (Figure 5C). Finally, the FFA and MAG level data were combined and the 99 samples were regrouped according to the predicted probability obtained by logistic regression (Figure 5D). The incidence of IPAH in the low-level group was reduced to 24.24%, and the incidence in the high-level group was 92.42%, 88.40% of the patients were included, indicating that the results were more sensitive and accurate when FFA and MAG were combined to predict the risk of IPAH.

## 3. Discussion

We included 69 IPAH patients and 30 healthy control subjects and performed a relative quantitative analysis of lipid content in this batch of 99 samples using LC–MS technology on a lipidomics platform. The lipids detected in this experiment comprised fourteen subclasses: FFA, MAG, DAG, TAG, LPA, LPS, PA, PG, PE, PI, PS, SM, SS, St, and a total of 588 lipid molecules. A series of single- and multi-dimensional statistical analyses processed the original data. The lipid subtype and lipid molecule levels were compared in IPAH patients after the onset of the disease, with healthy controls. TAG, LPS, PA, PG, PE, PI, PS, SS, St class lipid levels were significantly higher than that of healthy controls, indicating that these types of lipid molecules have a potential significance in the occurrence and development of IPAH disease.

The mean age of IPAH patients was (36.4 ± 10.0) years and female patients accounted for 79.7% of the patients, which suggests that IPAH is more prevalent in young and middle-aged female patients. The results was consistent with the demographic characteristics reported in the largest IPAH retrospective study in China (161 cases) [21]. The 187 cases of IPAH reported in the Scottish Registry study (1981–1989, 32 clinical centers) had a male-to-female ratio of 1:1.7 [22]. The different sex distribution of IPAH may be attributed to abnormal metabolism of sex hormones or to differences in the immunity systems between males and females.

The incidence of IPAH is higher in females than males but whether this is also reflected in sex differences in plasma lipid levels was uncertain. This study found no significant differences in the levels of lipid subclasses between male and female healthy controls or between male and female IPAH patients. However, TAG, St only showed significant differences between males, and the patient males were higher than healthy males PI; LPS, PG, PS only showed significantly different among the healthy females and IPAH females. Additionally, FFA, MAG, DAG, PA, PE, SS were significantly expressed between females and males. Studies provide evidence that there are gender differences in the distribution of lipids in populations and in pathologies [23,24,25]. Xuewen Wang et al. suggested that the network of hormone action might be an important regulator of lipid metabolism [24]. Martina Ambrož et al. suggested that there were significant differences in the lipid profile of type 2 diabetes (T2D) patients between males and females across their lifespan [26]. However, the lipid profile between males and females in the IPAH patients compared with healthy controls needs further study.

The outcome of logistic regression suggested that FFA, MAG, DAG, PE, SS, LPS, PA, PG, PI, PS and St could predict the incidence of IPAH. The multivariate logistic regression analysis revealed that FFA and MAG were independent predictors of IPAH. This finding is consistent with previous research suggesting that metabolic remodeling is present in IPAH [27]. This study found that MAG was an independent risk factor for IPAH. This is consistent with previous research indicating that MAG plays a significant role in cardiovascular disease and may serve as a potential therapeutic target [28], but its role in IPAH was unknown. Our results showed that MAG might be of significance in the development of IPAH. Consistent with previous studies [29,30], we also showed FFA was increase in IPAH. Previous reports, also found that FFA was associated with pulmonary hypertension [31]. High concentrations of circulating free fatty acids can lead to significant intracellular lipid accumulation, which can in turn trigger the production of reactive oxygen species and metabolic dysregulation. These processes culminate in cell death, inflammation, and tissue damage, which may contribute to the development and progression of various diseases, including cardiovascular disease [32,33]. Recent studies showed that circulating free fatty acids are increased by nearly twofold in patients with IPAH compared to healthy subjects, irrespective of other cardiovascular risk factors. These studies suggest that elevated levels of free fatty acids may play a significant role in the development and progression of IPAH [29,33]. Consistent with this, metabolic profiling of plasma from patients with IPAH showed that insulin resistance (IR) strongly correlates with altered lipid metabolic profiles. This further supports the notion that dysregulated lipid metabolism may contribute to the development and progression of IPAH [30,33]. The authors of this study found that, similar to atherosclerotic lesions in coronary artery disease (CAD), plexiform lesions in IPAH contain proinflammatory lipids, including oxidized low-density lipoprotein. These lipids may contribute to the recruitment of inflammatory cells and disruption of vascular cell function, which may promote the development and progression of plexiform lesions in IPAH [33]. The results showed increased FFA is a risk for developing IPAH. Currently, there are no reports regarding the effect of FFA on the incidence of IPAH and this requires further study.

The results of ROC analysis showed that individually, FFA and MAG had very good sensitivity. When FFA and MAG were combined there was high accuracy in predicting IPAH incidence. Particularly noteworthy was that in the samples of this study, all the patients with increased FFA and MAG levels were IPAH patients, indicating that the combined analysis of the two lipids can accurately predict the risk of IPAH. The ROC figure suggested that FFA was significant for predicting the prevalence of IPAH. We suggest that FFA and MAG might be the important lipid subclasses in the development of IPAH, but their mechanism of action in IPAH requires further investigation. 

This study has some limitations. The sample size of this study was small, with a limited number of patients and healthy controls, which could adversely affect the generalizability of the findings. Other potential confounding factors were medication use, comorbid conditions, and lifestyle factors, which may have influenced the results. Further studies with larger sample sizes and more comprehensive analyses are needed to confirm and expand upon the findings of this study. This study measured the lipid levels of patients when they were admitted to the hospital, and therefore changes in lipid levels during drug treatment were not assessed. Additionally, this study tested the changes in lipid content in various subgroups but further research is needed to identify the specific lipids that change in IPAH within the subclasses. Most important is to determine the underlying mechanisms whereby lipids play a role in the pathogenesis of IPAH.

## 4. Materials and Methods

### 4.1. Study Design and Subjects

This study included a total of 69 IPAH patients admitted to Shanghai Pulmonary Hospital from May 2013 to April 2019. Of these patients, 14 were male and 55 were female. The inclusion criteria specified a mean pulmonary artery pressure (mPAP) > 20 mmHg and pulmonary vascular resistance (PVR) < 3 wood units (WU), as per the 2022 (ESC/ERC) guidelines, as determined by right heart catheterization. Exclusion criteria were patients with pulmonary hypertension of known etiology, congenital left-to-right intracardiac shunts, portal hypertension, human immunodeficiency virus (HIV) infection, and patients who received hormone therapy (such as thyroid hormones, anabolic steroids, corticosteroids) in the past, or whose hormone production was suppressed by medication. Thirty age- and sex-matched healthy controls (6 males and 24 females) were also selected. Inclusion criteria were: Healthy, no previous history of other lung diseases and related illnesses, no history of related lung diseases in family members, and no drug or alcohol dependence.

This study adhered to the principles outlined in the Declaration of Helsinki and received approval from the Ethics Committee of the Shanghai Pulmonary Hospital (number: K20-195Y). All participants provided informed consent before their inclusion in this study. 

### 4.2. Clinical Data Collection

The diagnostic criteria for IPAH were in accordance with the “2022 ESC/ERS Guidelines for the diagnosis and treatment of pulmonary hypertension”. Patient data consisted of demographic information, 6 min walking distance (6MWD), World Health Organization functional class (WHO FC), and N-terminal fragment of pro-brain natriuretic peptide (NT-proBNP). Hemodynamic parameters included mean pulmonary arterial pressure (mPAP), mean pulmonary artery wedge pressure (mPAWP), mean right atrial pressure (mRAP), pulmonary vascular resistance (PVR), cardiac output (CO), and cardiac index (CI). Other laboratory parameters and treatment history were also recorded.

### 4.3. Blood Sample Collection

Blood samples were collected from all subjects in the morning after overnight fasting using an ethylenediaminetetraacetic acid anticoagulation tube. After standing for 30 min at 24 °C, the blood was centrifuged at 3500 rpm for 5 min in a 4 °C centrifuge and then the plasma layer was isolated. Lipids were extracted from plasma using dichloromethane extraction.

### 4.4. GC–MS Analysis

The liquid chromatography–mass spectrometry analysis was carried out using an EXION LC high-performance liquid chromatography in tandem with triple the quadrupole mass spectrometry system (HPLC-TripleQuad^TM^ 6500) as the instrument platform (AB SCIEX USA). The chromatography–mass spectrometry acquisition conditions were as follows: positive and negative ion detection mode, BEH amide HILIC column (100 mm × 2.1 mm i.d., 1.7 µm; Waters); mobile phase A was H_2_O/ACN (5:95, *v*/*v*, 10 mM ammonium acetate) and mobile phase B was H_2_O/ACN (50:50, *v*/*v*, 10 mM ammonium acetate). The flow rate used in the liquid chromatography–mass spectrometry analysis was 0.50 mL/min, with an injection volume of 5 μL and a column temperature of 40 °C. The sample mass spectrometry signals were obtained in both the positive ion (ESI+) and negative ion (ESI−) mode, and the data acquisition mode was MRM scanning.

### 4.5. Statistical Analysis

The table for normally distributed measures is presented as the mean ± standard deviation, while the table for non-normally distributed measures is shown as the median (interquartile range). Count data are presented as the sample size (percentage). Differences between the groups were assessed using the independent-samples *t*-test for normally distributed measures as well as the rank sum test for non-normally distributed measures. The statistical differences between multiple groups were analyzed using one-way ANOVA. The χ2 test was utilized to assess the between-group differences in count data. Predicted risk of morbidity was tested by one-way and multi-way logistic regression analysis. This study evaluated the ability of plasma lipid assessment to predict the increased risk of morbidity in patients using the receiver operator characteristic curve (ROC), which includes the area under the curve (AUC), sensitivity, specificity, and optimal cut-off value. A *p* value < 0.05 was considered to be statistically significant. Data were analyzed using SPSS 24 statistical software.

## 5. Conclusions

In conclusion, FFA and MAG lipid subclasses have potential as biomarkers for predicting IPAH. The high levels of FFA and MAG in plasma of IPAH patients suggest a potential role for FFA and MAG in the pathogenesis of IPAH. However, further studies are needed to determine whether FFA and MAG dysregulation are a cause or consequence of IPAH. Additionally, prospective studies are needed to determine whether targeting these lipid subclasses can be an effective therapeutic approach for the prevention and treatment of IPAH. Abnormal serum lipid distribution in patients with IPAH is an important element worthy of study, and its complex pathogenic mechanism remains unelucidated. In this study, we initially explored the potential relationship between lipid metabolites and IPAH. Therefore, our study suggested that IPAH patients exhibit a different distribution pattern of serum lipids, which may serve as a potential biomarker to aid in clinical diagnosis.

## Figures and Tables

**Figure 1 ijms-24-14280-f001:**
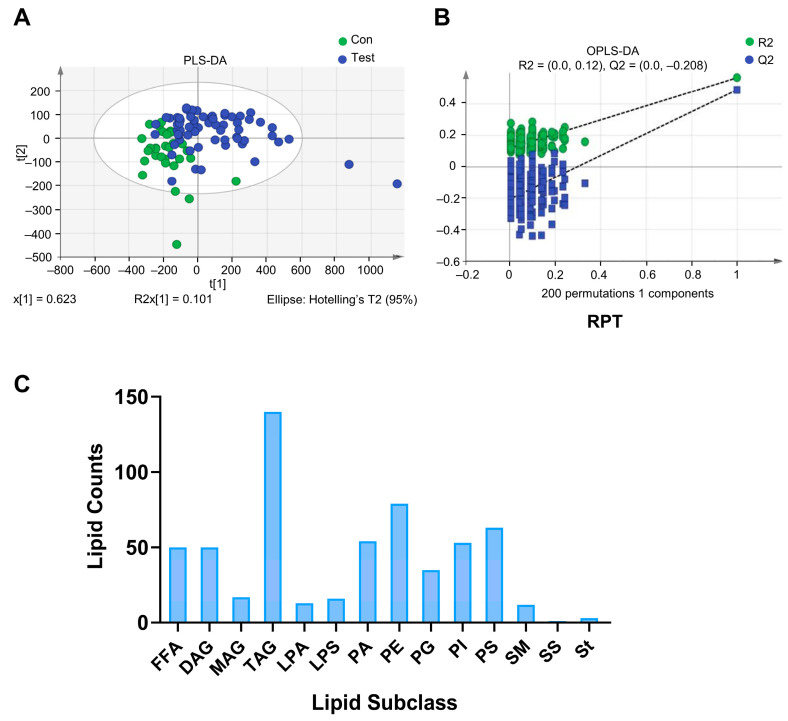
Initial data processing for LC–MS. (**A**) PLS-DA analysis of lipid levels in IPAH patients and healthy controls. (**B**) RPT evaluation of lipid levels in IPAH patients and healthy controls. (**C**) The number of lipid species in various lipid classes. FFA: free fatty acid, MAG: monoacylglycerol, DAG: diacylglycerol, TAG: triacylglycerol, LPA: lysophosphatidic acid, LPS: lysophosphatidylserine, PA: phosphatidic acid, PE: phosphatidyl ethanolamine, PG: phosphatidyl glycerol, PI: phosphatidylinositol, PS: phosphatidylserine, SM: sphingomyelin, SS: sphingosine, and St: sterol.

**Figure 2 ijms-24-14280-f002:**
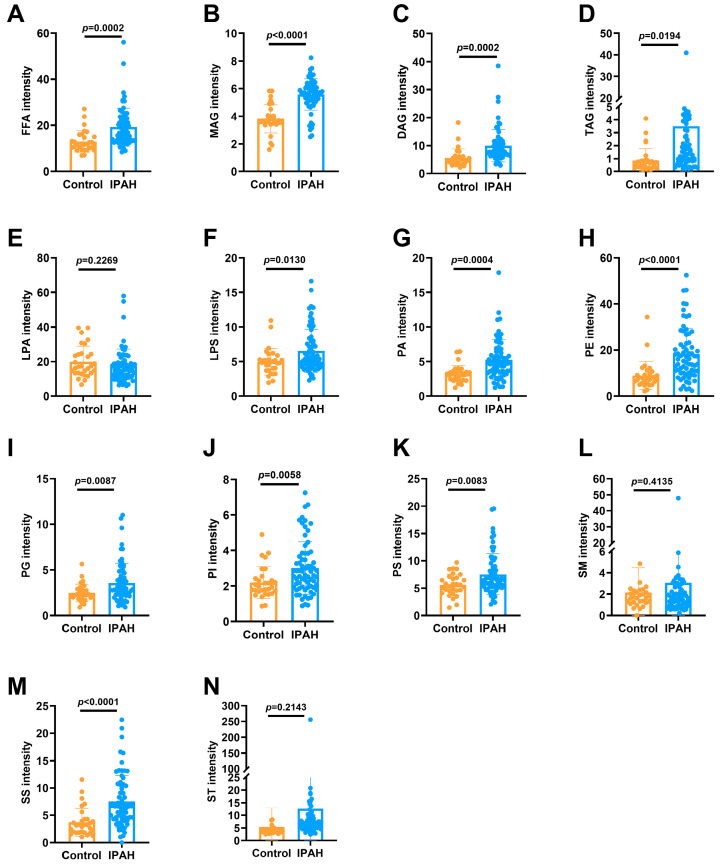
Levels of the 14 lipid subclasses in IPAH patients and healthy controls. (**A**–**N**) Relative intensity of the 14 (FFA, MAG, DAG, TAG, LPA, LPS, PA, PE, PG, PI, PS, SM, SS, ST) lipid subclasses in IPAH patients and healthy controls. FFA: free fatty acid, MAG: monoacylglycerol, DAG: diacylglycerol, TAG: triacylglycerol, LPA: lysophosphatidic acid, LPS: lysophosphatidylserine, PA: phosphatidic acid, PE: phosphatidyl ethanolamine, PG: phosphatidyl glycerol, PI: phosphatidylinositol, PS: phosphatidylserine, SM: sphingomyelin, SS: sphingosine, and ST: sterol.

**Figure 3 ijms-24-14280-f003:**
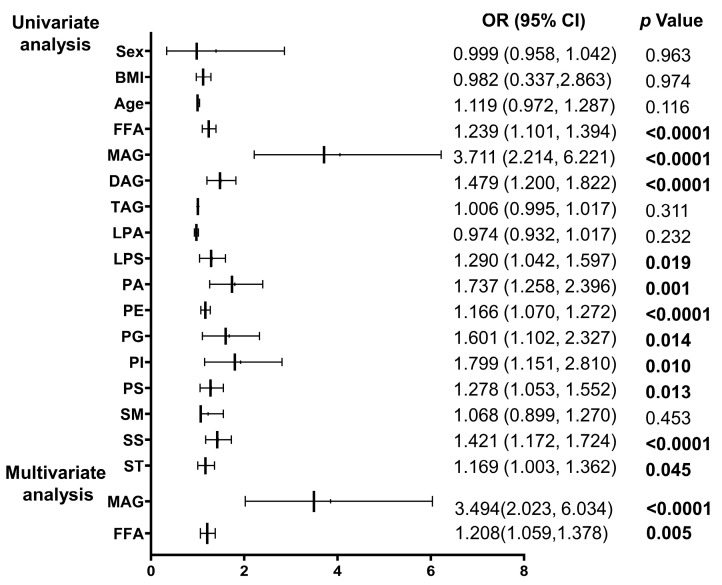
Logistic regression analysis of lipids. FFA: free fatty acid, MAG: monoacylglycerol, DAG: diacylglycerol, TAG: triacylglycerol, LPA: lysophosphatidic acid, LPS: lysophosphatidylserine, PA: phosphatidic acid, PE: phosphatidyl ethanolamine, PG: phosphatidyl glycerol, PI: phosphatidylinositol, PS: phosphatidylserine, SM: sphingomyelin, SS: sphingosine, St: sterol, OR: odds ratio, and 95% CI: 95% confidence interval.

**Figure 4 ijms-24-14280-f004:**
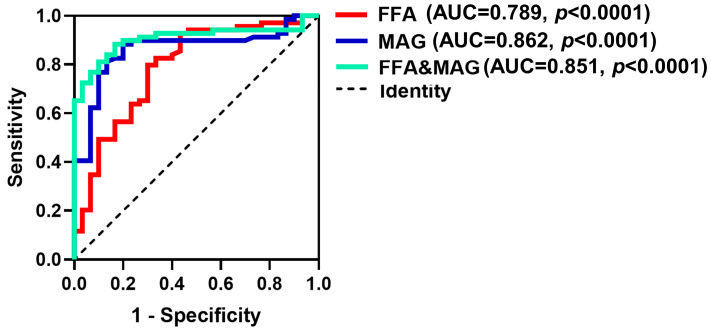
ROC analysis of FFA, MAG and their combined detection to predict IPAH. FFA: free fatty acid, MAG: monoacylglycerol, FFA and MAG, FFA and MAG combined prediction probability based on logistic regression; AUC, area under the curve.

**Figure 5 ijms-24-14280-f005:**
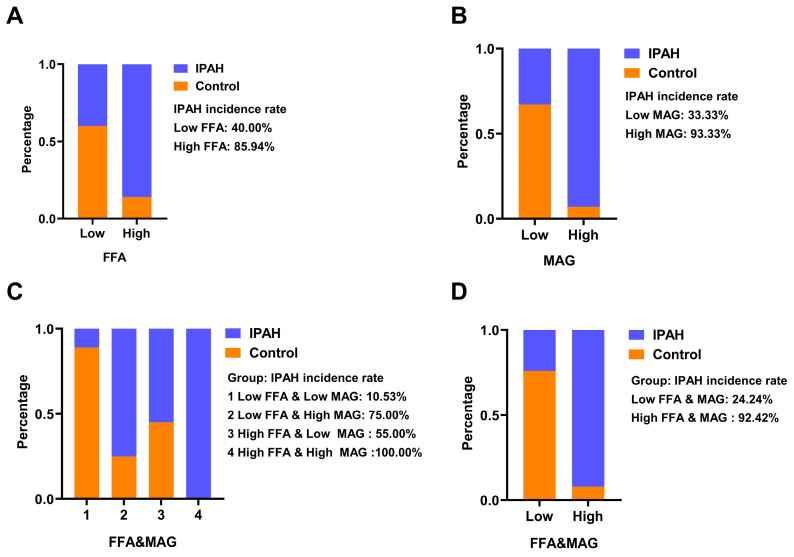
Sample grouping based on FFA and MAG lipid levels and sample grouping based on FFA and MAG joint prediction probability. (**A**) Sample grouping based on FFA lipid levels. (**B**) Sample grouping based on MAG lipid levels. (**C**) Sample grouping based on FFA and MAG lipid levels. FFA and MAG combined group; Chi-square test. (**D**) Sample grouping of FFA and MAG joint prediction probability. FFA and MAG, FFA and MAG jointly predict the probability group. Chi-square test.

**Table 1 ijms-24-14280-t001:** Baseline characteristics of patients.

Characteristics	Control (*n* = 30)	IPAH (*n* = 69)	*p* Value
Age, years	36.5 ± 10.7	36.4 ± 10.0	0.149
Female, n (%)	24 (80.0)	55 (79.7)	0.611
BMI, kg/m^2^	21.6 ± 2.7	22.8 ± 3.4	0.458
WHO-FC III/IV (%)	N/A	41 (59.4)	N/A
6MWD, m	N/A	393.9 ± 104.8	N/A
Death, n (%)	N/A	9 (13)	N/A
Hemodynamics			
mRAP, mmHg	N/A	3.5 (1.0–6.0)	N/A
mPAP, mmHg	N/A	55.0 (49.0–62.5)	N/A
mPAWP, mmHg	N/A	6.0 (3.0–8.0)	N/A
PVR, wood units	N/A	11.7 ± 6.0	N/A
CO, L/min	N/A	4.2 (3.5–5.2)	N/A
CI, L/min/m^2^	N/A	2.6 (2.2–3.7)	N/A
**Laboratory**			
NT-proBNP, pg/mL	N/A	1019.9 ± 926.3	N/A
TBIL, µmol/L	N/A	15.0 (11.0–20.4)	N/A
DBIL, µmol/L	N/A	5.0 (4.0–7.4)	N/A
UA, mg/dL	N/A	410.1 ± 116.8	N/A
CR, µmol/L	N/A	63.3 ± 13.0	N/A
GLU, mmol/L	N/A	4.8 (4.4–5.1)	N/A
TC, mmol/L	N/A	4.0 ± 1.0	N/A
TG, mmol/L	N/A	1.4 ± 0.7	N/A
HDL, mmol/L	N/A	1.0 ± 0.3	N/A
LDL, mmol/L	N/A	2.4 (2.0–3.1)	N/A
HGB, g/L	N/A	141.8 ± 18.6	N/A
RBC, 10^12^/L	N/A	4.8 ± 0.6	N/A
WBC, 10^9^/L	N/A	6.2 (5.3–7.7)	N/A
**Specific therapy**			
PDE-5 inhibitors	N/A	15 (21.7)	N/A
ERAs	N/A	6 (8.7)	N/A
Prostacyclin analogs	N/A	0 (0)	N/A
Combination	N/A	41 (59.4)	N/A
Non-specific medication	N/A	7 (10.1)	N/A

The data were expressed as the mean ± standard deviation, median and quaternary interval. 6MWD = 6 min walking distance, BMI = body mass index, CI = cardiac index, CO = cardiac output, CR = creatinine, DBIL = direct bilirubin, ERA = endothelin receptor antagonist, GLU = glucose, HDL = high-density lipoprotein, HGB = hemoglobin, LDL = low-density lipoprotein, mPAP = mean pulmonary arterial pressure, mPAWP = mean pulmonary artery wedge pressure, mRAP = mean right atrial pressure, NT-proBNP = N-terminal pro-brain natriuretic peptide, PDE-5 = phosphodiesterase 5, 5, PVR = pulmonary vascular resistance, RBC = red blood cell, TBIL = total bilirubin, TC = total cholesterol, TG = triglyceride, UA = uric acid, WBC = white blood cell, WHO FC = World Health Organization functional classification, and *p* value in the table are control vs. IPAH.

**Table 2 ijms-24-14280-t002:** Levels of lipid subclasses between males and females.

Lipid Intensity	Control (*n* = 30)		*p* Value	IPAH (*n* = 69)		*p* Value
	Male (*n* = 6)	Female (*n* = 24)		Male (*n* = 14)	Female (*n* = 55)	
FFA	12.0 ± 2.3	13.3 ± 5.0	0.553	18.4 ± 7.1 *	19.5 ± 8.5 ^#^	0.650
MAG	4.0 ± 0.8	3.8 ± 1.0	0.690	5.6 ± 0.9 *	5.6 ± 1.2 ^#^	0.954
DAG	4.7 ± 2.6	5.8 ± 3.4	0.450	10.4± 5.9 *	9.9 ± 5.8 ^#^	0.766
TAG	7.6 ± 8.3	23.5 ± 72.8	0.603	25.1 ± 23.7 *	37.6 ± 66.5	0.492
LPA	21.2 ± 8.2	19.7 ± 9.3	0.717	15.4 ± 6.4	18.0 ± 10.4	0.381
LPS	4.8 ± 1.8	5.0 ± 2.0	0.852	7.1 ± 3.7	6.4 ± 3.0 ^#^	0.456
PA	2.8 ± 0.7	3.3 ± 1.3	0.335	4.8 ± 2.4 *	5.4 ± 73.0 ^#^	0.482
PE	6.3 ± 2.4	9.7 ± 6.6	0.223	15.9 ± 8.7 *	18.8 ± 11.8 ^#^	0.386
PG	2.4 ± 0.7	2.5 ± 1.0	0.780	2.9 ± 1.5	3.7 ± 2.3 ^#^	0.214
PI	2.6 ± 1.0	2.1 ± 0.9	0.272	2.4 ± 0.9	3.2 ± 1.6 ^#^	0.089
PS	6.1 ± 2.0	5.5 ± 2.0	0.511	6.2 ± 2.3	7.9 ± 4.0 ^#^	0.121
SM	1.6 ± 0.7	2.3 ± 2.6	0.550	1.7 ± 1.1	3.4 ± 6.6	0.346
SS	2.2 ± 0.8	4.1 ± 2.7	0.113	7.2 ± 4.3 *	7.6± 4.8 ^#^	0.728
St	3.9 ± 1.3	5.8 ± 8.3	0.590	6.9 ± 2.4 *	14.1 ± 34.8	0.444

Data are presented as the mean ± standard deviation. FFA: free fatty acid, MAG: monoacylglycerol, DAG: diacylglycerol, TAG: triacylglycerol, LPA: lysophosphatidic acid, LPS: lysophosphatidylserine, PA: phosphatidic acid, PE: phosphatidyl ethanolamine, PG: phosphatidyl glycerol, PI: phosphatidylinositol, PS: phosphatidylserine, SM: sphingomyelin, SS: sphingosine, St: sterol, * *p* < 0.05 (Control male vs. IPAH male); ^#^ *p* < 0.05 (Control female vs. IPAH female); *p* values in the table are control male vs. control female, IPAH male vs. IPAH female.

## Data Availability

Data will be available from the corresponding author under reasonable request.

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
