# Peer review of "The Impact of Abnormal Lipid Metabolism on the Occurrence Risk of Idiopathic Pulmonary Arterial Hypertension"

_ijms, 2023, doi:10.3390/ijms241814280_

Round 1

Reviewer 1 Report

The manuscript is interesting. The topic "The impact of abnormal lipid metabolism on the occurrence risk of idiopathic pulmonary arterial hypertension" is hot. The manuscript is quite well written. However,I have some suggestions:

1) Abstract. To determine whether lipid molecules can be used as potential biomarkers for idiopathic pulmonary arterial hypertension (IPAH), providing important reference value for early diagnosis and treatment. Please, add a brief background on IPAH.

2) Abstract. Among the lipid subclasses, FFA and MAG have potential as biomarkers for predicting the pathogenesis of IPAH, which may ben-efit the prognosis of patients and improve the early diagnosis of IPAH. Please, improve the conclusions.

3) 1. Introduction L35-37. Idiopathic pulmonary arterial hypertension (IPAH) is a progressive disease that im-  pacts the precapillary pulmonary vasculature, but the specific risk factors that contribute  to IPAH remain unknown[1]. Please, improve the paragraph on the PAH and add some references, such as:

a- Int. J. Mol. Sci. 202324, 8462. https://doi.org/10.3390/ijms24108462. Pulmonary Hypertension: New Insights and Recent Advances from Basic Science to Translational Approaches. 

b- Diagnostics (Basel). 2022 Mar 1;12(3):616. doi: 10.3390/diagnostics12030616. An Overview of Different Techniques for Improving the Treatment of Pulmonary Hypertension Secondary in Systemic Sclerosis Patients. 

4)  1. Introduction L70-73.  The objective of this study was to determine the levels of lipid molecules in the  plasma of individuals with IPAH and healthy individuals using liquid chromatography- mass spectrometry. Additionally, the study aimed to evaluate the potential of lipid mole- cules as biomarkers to predict IPAH and to determine the possible role of lipids in the  pathogenesis of IPAH. Please, improve the description of study aim.

5) 2. Results. Please, underline in the manuscript the most important results and statistical values to support the data. 

6) 3. Discussion L222-226. We included 69 IPAH patients and 30 healthy control subjects. The mean age of IPAH  patients was 36.4 ± 10.0 years, female patients accounted for 79.7%, which is a higher pro- portion than male patients, suggesting that IPAH tends to occur in young and middle- aged female patients, which is consistent with the demographic characteristics reported in the largest IPAH retrospective study in China (161 cases)[15]. Please, rewrite this sentence and summarise the most important results of the study.

7) 3. Discussion. Undeline the possible limitations of the study

8) 5. Conclusions L365-371. In conclusion, FFA and MAG lipid subclasses have potential as biomarkers for pre-dicting IPAH. The high levels of FFA and MAG in plasma of IPAH patients suggests a potential role for FFA and MAG in the pathogenesis of IPAH. However, further studies  are needed to determine whether FFA and MAG dysregulation are a cause or consequence of IPAH. Additionally, prospective studies are needed to determine whether targeting  these lipid subclasses can be an effective therapeutic approach for the prevention and  treatment of IPAH. Please, underline the possible clinical implication of the study.

Reviewer 2 Report

Dear Authors,

Please, take these suggestions into consideration:

1)    ABSTRACT.

Please, divide the abstract into correct sections for a research article (i.e. background, methods, results, ..).

2)    ABSTRACT. Lines 21-23. “In this study, lipid profiling was performed on plasma samples from 69 IPAH patients and 30 healthy controls to compare the levels of lipid molecules in the four groups of patients,...”

Please, correct the contradiction within this sentence: you state that the subjects enrolled in this study were divided into IPAH patients and healthy controls, but then you talk about four (4) groups of patients. Did you mean two (2) or are there actually four subgroups? If so, please specify the characteristics of these subgroups.

3)    INTRODUCTION. Line 63. “However, few reports investigate the relationship between lipid metabolism and IPAH”.

Since you stated that there are a few reports which investigate the relationship between lipid metabolism and IPAH but you only cited one (Zhao, Q.H.; Peng, F.H.; Wei, H.; He, J.; Chen, F.D.; Di, R.M.; Jiang, X.; Jiang, R.; Chen, Y.J.; Heresi, G.A.; et al. Serum high-density lipoprotein cholesterol levels as a prognostic indicator in patients with idiopathic pulmonary arterial hypertension. Am J Cardiol 2012, 110, 433-439, doi:10.1016/j.amjcard.2012.03.042.), please add some more references, briefly summarizing their results.

4)    TABLE 1.

Please, add laboratory data of the control group.

Dear Authors,

the quality of English in this work is fine, I detected only a few grammatical errors, which only require a quick revision.

Reviewer 3 Report

The authors examined diagnostic capacity of lipids for IPAH. The study is well designed  but has a limitied sample size which might have affected statistical analyses.

The title of the study is not optimal; I would suggest: Assiociation of plasma lipids with risk for IPAH; or Increased plasma FFA and MAG are strongly associated with incidence of IPAH.

My major criticism is a very poor data presentation:

lane 78: Body mass index should be body mass index

lane 82: have been listed - are listed

lane 96: The farher awy....this sentence should be re-written.

lanes 106, 107: Sphingosine - sphingosine, Sterol - sterol

lane 112: The type of...should be replaced with: number of lipid species in various lipid classes.

The authors should provide as a supplemental table all measured lipid species with concentrations and explain in the methods how FFA, MAG values used for statistical analyses were obtained.

The authors should report on the number of missing values or those under detection limit.

How was distribution of the data in particular FFA and MAG; would log transformation affect results of logistic regression analyses?

Do OR (95% CI) presented as increase per 1 unit?

lane 119: ,...we found that the lipid levels...please delete ''lipid''

lane 120: replace ''demonstrated a...'' with ''were'' and delete ''trend'' in the same sentence.

lane 122, 123: Replace the sentence with: ''LPA and SM were similar in both groups.''

lane 128: Figure 2. Differences in lipid subclasses between IPAH patients and healthy controls.

Please include results described in the lanes 138 - 150 into Fig. 2; accordingly, each subfigure of Fig. 2 should contain female, male, all; alternatively show the data described in the lanes 138-150 as  Table 2.

Please shift the present Table 2 into supplement.

lane 158: Logistic regression analyses of IPAH for various lipid subclasses

Please expand Fig. 3 and show in the multivariable analysis OR and 95% CI for  for all variable sused in the univariable analysis.

Please change in Fig. 3 bold for OR and 95% CI into normal font but leave in bold significant P-values.

lane 186; please present AUC as 0.789 etc.. and not as %.

lane 187, 188: Please re-writte the sentence.

lanes 195 - 214: very poor description of data; please simplify and improve description of the data presented in Fig. 5

Discussion contains too much repetiton of the results and too less implications of the results.

English grammar and syntax as well as overall improvement of data presentation is need.

Reviewer 4 Report

This manuscript describes a case-control study in which 69 patients with idiopathic pulmonary arterial hypertension (IPAH) were enrolled, and odd ratios and logistic regression analysis were calculated to explore the potential role of lipids in IPAH. Outcomes showed, out of 14 lipids tested, 12 lipid levels were higher in IPAH patients than in healthy controls, with more prominence in free fatty acids and monoacylglycerol. This study focuses on the broad lipid profile in patients with IPAH, which brings some novelty. The stratification according to gender is decent. I list several major concerns for the authors which are needed to be addressed.   

 1.    The size of the sample is too small, while the patients with/without IPAH were not matched accordingly to minimum unrelated factors.

2.    Simple statistical findings between two variables cannot build a convincing relationship between phenomena. Lipid metabolism does not directly influence the risk of IPAH. Please upload your original database to show the intact background.

3.    How can you rule out confounding factors such as obesity and obstructive sleep apnea, which are known as risk factors of IPAH?

4. What are the medical or surgical therapies for your patients?

Round 2

Reviewer 1 Report

The authors have addressed my questions adequatly. They have modified the manuscript and taken into account all suggestions. In my opinion, this has improved the manuscript. I have no further comments.  

Author Response

Thank you for your hard work and kind patience.

Reviewer 3 Report

The authors significantly improved the manuscript.

I'm still not sure that the title is an optimal one, buit its the authors' paper anad they do not need to follow my suggestion in this point.

lane 184: ...showed please add THAT FFA  ( that is missing presently)

fine, only some minors observed

Reviewer 4 Report

This manuscript describes a case-control study in which 69 patients with idiopathic pulmonary arterial hypertension (IPAH) were enrolled, and odd ratios and logistic regression analysis were calculated to explore the potential role of lipids in IPAH. Outcomes showed, out of 14 lipids tested, 12 lipid levels were higher in IPAH patients than in healthy controls, with more prominence in free fatty acids and monoacylglycerol. This study focuses on the broad lipid profile in patients with IPAH, which brings some novelty. The stratification according to gender is decent. The authors responded to my questions and made detailed revisions. After explanations, the conclusion is less vulnerable. Overall, the manuscript and figures are carefully arranged and logically organized. However, I did not see the raw data after data masking. Please address this issue for further consideration.
